# Systematic Review of Single-Agent vs. Multi-Agent Chemotherapy for Advanced Pancreatic Adenocarcinoma in Elderly vs. Younger Patients

**DOI:** 10.3390/cancers15082289

**Published:** 2023-04-13

**Authors:** Alison Lewis, Adnan Nagrial

**Affiliations:** 1School of Medicine, The University of Sydney, Camperdown, NSW 2006, Australia; 2Westmead Hospital, Westmead, NSW 2145, Australia

**Keywords:** pancreatic adenocarcinoma, chemotherapy, advanced neoplasms, advanced care, multi-agent chemotherapy

## Abstract

**Simple Summary:**

Pancreatic adenocarcinoma is an aggressive tumour with a high mortality, often diagnosed at advanced stages. The current standard of care for first-line therapy is multi-agent FOLFIRINOX or gemcitabine plus nab-Paclitaxel. Pancreatic adenocarcinoma predominantly effects the elderly population, albeit these patients are underrepresented in clinical trials. This systematic review aims to review the efficacy of multi-agent versus single-agent chemotherapy for advanced pancreatic adenocarcinoma in elderly versus young populations.

**Abstract:**

Purpose: To systematically review all studies comparing multi-agent to single-agent chemotherapy in the first and second-line setting for unresectable pancreatic adenocarcinoma, so as to compare the outcomes of young and elderly patients. Methods: This review searched three databases for relevant studies. The inclusion criteria were diagnosis of locally advanced or metastatic pancreatic adenocarcinoma, comparison of an elderly versus young population, comparison of single-agent versus multi-agent chemotherapy, data on survival outcomes, and randomised controlled trials. The exclusion criteria were phase I trials, incomplete studies, retrospective analyses, systematic reviews, and case reports. A meta-analysis was performed on second-line chemotherapy in elderly patients. Results: Six articles were included in this systematic review. Three of these studies explored first-line treatment and three explored second-line treatment. In the subgroup analysis, the meta-analysis showed statistically improved overall survival for elderly patients receiving single-agent second-line treatment. Conclusions: This systematic review confirmed that combination chemotherapy improved survival in the first-line treatment of advanced pancreatic adenocarcinoma, regardless of age. The benefit of combination chemotherapy in second-line studies for elderly patients with advanced pancreas cancer was less clear.

## 1. Introduction

Pancreatic adenocarcinoma is an aggressive and highly lethal tumour with a five-year survival rate of 8% [1,2]. If identified early, surgical resection can potentially be curative [1]. However, effective screening is not currently available and pancreatic adenocarcinoma is often diagnosed at advanced stages [3].

Consequently, palliative chemotherapy is an important therapy for most patients diagnosed with unresectable pancreatic cancer. The current standard for first-line treatment is gemcitabine plus nab-Paclitaxel or FOLFIRINOX, both of which are multi-agent therapies [3]. There is currently no universal agent recommended for second-line treatment. In Asia, S-1 is commonly used for gemcitabine-refractory pancreatic adenocarcinoma [4]. Gemcitabine plus nab-Paclitaxel, FOLFIRINOX, and FOLFOX are options as second-line therapies globally [5].

Pancreatic cancer predominantly affects older patients, with the median age of diagnosis being 71 years [6]. Clinical trials do not always reflect this distribution and elderly patients are underrepresented [6]. This can be due to comorbidities, socioeconomic factors, and other exclusion criteria such as age [7]. In the absence of robust evidence, there is uncertainty about the efficacy and safety of chemotherapy in older patients, with one study showing that chemotherapy management plans can be implemented in older patients, without a significant variation in results [8]. Other studies have shown that elderly patients are not appropriate candidates for some chemotherapy regimens, such as FOLFIRINOX, due to severe toxicity and a poor prognosis [9].

The aim of this study was to systematically review all of the studies comparing multi-agent to single-agent chemotherapy in the first and second-line setting for unresectable pancreatic adenocarcinoma, so as to compare the outcomes of young and elderly patients.

## 2. Materials and Methods

Search strategy: A protocol was written and submitted to PROSPERO [10]. The PROSPERO registration number was CRD42021245088. Literature searches were conducted in Medline, Embase, and Cochrane Central using the terms shown in Appendix A. The search was limited to results in the English language. As the first study demonstrating the benefit of chemotherapy in pancreas cancer was not published until 1997, clinical trials between 1990–2020 were included [1]. The authors were not contacted for raw data. Reference lists of the selected trials were scanned for any other relevant trials.

Selection criteria: Studies were included in the review if they met all of the following criteria: randomised controlled trials, patients had a histological diagnosis of locally advanced or metastatic pancreatic adenocarcinoma, compared a single-agent versus multi-agent chemotherapy in pancreatic adenocarcinoma, compared outcomes in an older versus younger population, and provided data on overall survival. The following studies were excluded: phase I trials, incomplete studies, retrospective analyses, systematic reviews, and case reports. Two independent reviewers (A.L. and A.N.) screened the results of the search strategy by title and subsequently by abstract. The full-text articles of potentially eligible studies were then independently assessed by the same two authors against the inclusion criteria, and any disagreement was resolved by consensus.

Data extraction and quality assessment: Two independent reviewers (A.L. and A.N.) extracted the data from the included studies using a pre-determined collection sheet according to the Preferred Reporting Items for Systematic Reviews and Meta-Analyses Statement [11]. Full-text articles were reviewed and reasons for exclusion were recorded. For any studies where consensus could not be reached, a third investigator was available to provide input. Processed data were compiled using Microsoft Excel. Data items included study title, authors, year of publication, country, publication journal, funding source, eligibility criteria, participant demographics (age, sex, and stage of disease), sample size, risk of bias, details of drug regimens, and outcomes (overall and by subgroup). Risk of bias was assessed looking at three domains: performance, selection, and blinding of outcome assessment bias. This assessment was based on the Cochrane risk-of-bias tool [12].

End points: The primary outcome was overall survival (OS) for single-agent versus multi-agent chemotherapy. Secondary outcomes included progression free survival (PFS), overall response rate (ORR), and adverse events (nausea, vomiting, fatigue, and neutropenia). Data on Grade 3 or 4 adverse events were extracted if available. Primary and secondary outcomes were evaluated again with a subgroup analysis looking at elderly versus younger populations. Descriptive statistics for each study have been provided. Statistical analysis of the overall hazard ratio (HR) for OS was calculated using Revman Version 5.4 [13]. A statistical test with a *p*-value of <0.05 was considered significant. A HR > 1 reflects worse survival with multi-agent chemotherapy compared with single-agent chemotherapy. The I^2^ statistic was calculated to evaluate the extent of variability attributable to statistical heterogeneity between trials. It was considered statistically significant when heterogeneity was <0.10 or I^2^ > 25%. Data were analysed using a random effects model using the inverse variation method. Sensitivity analyses were also conducted to assess the influence of each study on overall estimate of HR by sequential removal of individual studies. All *p*-values were two-sided. HREC approval was not required for this systematic review of published literature.

## 3. Results

The initial search strategy was conducted on 11 December 2020 and found 1297 articles. After screening duplicates, titles, and abstracts, there were 65 results. These 65 trials were analysed for eligibility by evaluation of the full text. Of these, 59 were excluded because they did not meet the inclusion criteria, as listed in Figure 1. Therefore, six studies were included in this systematic review.

### 3.1. Characteristics of Included Trials

Six eligible trials were identified and have been described in Table 1, with overall outcomes in Table 2. Of these, four studies provided number of patients per treatment arm in the age subgroup analysis. The elderly patient subgroup comprised a smaller sample size than younger patients in all six studies. The median age ranged from 61–65 across the six studies.

The total number of included patients was 3048, where 1665 patients were allocated as single-agent therapy and 1383 were allocated as multi-agent therapy. Of the included studies, three analysed first-line therapy [15,16,17] and three analysed second-line therapy [4,18,19]. Four of the trials were conducted in Japan, and all had been published. All studies reported ORR, PFS, and OS.

Of the first-line studies, the median OS was 6.6–8.8 months in the single-agent subgroup and 8.7–11.1 months in the multi-agent subgroup. The single-agent used was gemcitabine in all the studies. Two of the three studies showed statistically significant prolonged OS in multi-agent treatment arms [15,16]. Unlike the other two studies, Imaoka et al. (2016) included an Asian population and showed numerically improved OS albeit this was not statistically significant [17]. All three studies showed a statistically significant improvement in PFS for multi-agent vs. single-agent chemotherapy.

Of the second-line studies, the median OS was 5.8–7.9 months in the single-agent subgroup and 6.8–7.6 months in the multi-agent subgroup. None of the second-line studies demonstrated a statistically significant difference in OS. All three studies were in an Asian population and used S-1 as the single-agent. The PFS of multi-agent chemotherapy was statistically increased compared with single-agent treatment in Ioka et al., 2019 [19].

Multi-agent therapy resulted in numerically improved response rates in both the first-line and second-line setting.

### 3.2. Subgroup Analysis

A subgroup analysis of single-agent vs. multi-agent treatment in elderly and younger populations was then performed, shown in Table 3 and Table 4. In the subgroup analysis, all studies provided OS data while only one study provided PFS and ORR data. Four studies used age 65 as the age cut-off [4,15,16,18] while two used 70 as the age cut-off [17,19] to stratify by young vs. old.

In the single-agent arms, the median OS ranged from 6.8–10 months in the younger patients and 6.5–8.5 months in the older patients. In the multi-agent arm, the median OS ranged from 9.6–10.2 months in the younger patients and 7.7–10.2 months in the older patients.

Of the first-line studies, Conroy et al. (2011) and Goldstein et al. (2015) showed statistical significance favouring prolonged OS for multi-agent chemotherapy in both elderly and young subgroups [16]. This was confirmed on the meta-analysis of first-line studies comparing single vs. multi-agent chemotherapy in both elderly and young patients (Figure 2a,b).

Of the second-line studies, OS favoured monotherapy in elderly patients (HR > 1) and combination therapy in younger patients (HR < 1). However, only Ohkawa et al. (2015) found a statistically significant difference. PFS also favoured monotherapy in elderly patients and combination chemotherapy in younger patients, although only Ohkawa et al. (2015) demonstrated a statistically significant difference in the younger patient subgroup. No data were reported for ORR in all second-line studies in the subgroup analysis.

A meta-analysis of all second-line studies comparing multi-agent chemotherapy to single-agent (S1) in each patient population was performed. Summarised in Figure 3a,b, the elderly population demonstrated a pooled HR for OS of 1.38 [1.08–1.75: *p* = 0.009] compared with 0.74 [0.56–0.98: *p* = 0.03] for younger patients. This suggests a potential superiority of single-agent second-line chemotherapy in elderly patients in these included studies.

### 3.3. Toxicity

Five studies reported overall adverse events shown in Table 5, and Imaoka et al. (2016) reported adverse events by subgroup, as summarised in Table 6. The absolute adverse events seen in the combinations arms of the studies were greater than the single-agent arms. Imaoka et al. (2016) showed that elderly patients receiving first-line single-agent chemotherapy reported a higher incidence of neutropenia, nausea, vomiting, and fatigue than younger patients receiving the same treatment [17]. However, elderly patients had a lower incidence of nausea and vomiting in the multi-agent chemotherapy treatment arm compared with younger patients receiving multi-agent chemotherapy [16].

### 3.4. Assessment of Bias

All of the studies had a low overall risk of bias. All six studies had a low risk of selection bias as they were randomised. All of the studies had a high risk of performance bias as they were open-label, and two had a high risk of outcome assessment bias as their primary outcome was PFS. This is summarised in Appendix B.

## 4. Discussion

Single-agent gemcitabine has historically been the gold standard for first-line treatment of advanced pancreatic adenocarcinoma [20]. More recently, combination treatment with gemcitabine plus nab-Paclitaxel or FOLFIRINOX has been associated with a statistical benefit in OS, PFS, and ORR for eligible patients enrolled in the studies receiving first chemotherapy for unresectable/metastatic pancreatic ductal adenocarcinoma [21]. We found that when stratified by age, both elderly and young patients had better outcomes with combination chemotherapy. Interestingly, elderly patients had a lower incidence of nausea, vomiting, and fatigue with combination treatment than younger patients for first-line therapy in the single study that provided this data.

Second-line therapy for advanced pancreatic adenocarcinoma has less effective options available for patients compared with first-line therapy [22]. The overall outcomes revealed a similar OS between single-agent and doublet therapy in the second-line studies. However, the subgroup analysis revealed that elderly patients may not have gained a benefit from multi-agent chemotherapy, while younger patients experienced better outcomes with combination chemotherapy. Toxicity trended towards being higher in multi-agent treatment arms compared with single-agent treatment arms in second-line therapy studies; however, subgroup analysis by age was not available.

Recently, a retrospective cohort study that involved 116 elderly patients (age range being 75–84 years) with advanced pancreatic adenocarcinoma showed that first-line multi-agent chemotherapy using gemcitabine plus nab-Paclitaxel was beneficial for OS [23]. The median OS was 21.8 months for locally advanced and 13.3 months for metastatic disease [23]. These results were comparable with younger populations [23]. However, the frequency of toxic events was higher in elderly patients requiring more dose reductions and delays in treatment [23]. Another review found that gemcitabine plus nab-Paclitaxel may be more appropriate in an elderly population than FOLFIRINOX due to the side effect profile [8]. These results were in line with the findings in this review, which supported the treatment of elderly patients with multi-agent chemotherapy for first-line treatment.

A systematic review of second-line chemotherapy by Nagrial et al. (2015) demonstrated an increased OS compared with supportive care alone [24]. This was supported by another review on advanced pancreatic adenocarcinoma that confirmed patients benefit from second-line chemotherapy [8]. Neither study recommended a specific agent, which notably highlighted the low utilisation of second-line treatment largely due to low performance status [24]. The authors concluded that a speedier delivery of second-line treatment could improve the overall outcomes for patients [24]. Nagrial et al. (2015) found that combination treatment was correlated with a higher OS than single-agent treatment, although it did not provide breakdown by age [24].

Of the second-line studies included in the current review, all three assessed an Asian population using S-1 as the single-agent. S-1 is commonly used in Japan, South Korea, and Taiwan for the treatment of gemcitabine-refractory pancreatic adenocarcinoma [4,25]. S-1 results in similar levels of its active ingredient fluorouracil, although it is correlated with more significant gastrointestinal toxicity in Caucasian populations [17]. The findings thus cannot be extrapolated to Western patients. The meta-analysis in Figure 3a showed that single-agent therapy with S-1 had improved OS compared with multi-agent chemotherapy in elderly patients receiving second-line treatment. This may be due to greater adverse effects from the multi-agent chemotherapy requiring dose reductions in the elderly [26,27].

There is a scarcity of research in multi-agent versus single-agent chemotherapy in elderly populations, especially for second-line treatment [28]. Overall, of the 54 potential RCTs, only 6 provided data on outcome stratified by age. The subgroup analysis of this paper aimed to explore this gap and provide guidance to clinicians on the management of elderly patients. The databases used were comprehensive and reference lists were also screened [29].

Despite this, there was a small sample size of elderly patient subgroups included in this review. The available trials had a lack of data on elderly patients. Most available literature detailing the treatment outcomes of elderly patients with advanced pancreatic adenocarcinoma is in the form of retrospective studies [6].

The weaknesses of this review included the lack of availability of all data from the included studies. None of the studies had all of the endpoints required by this systematic review, which limited inclusion. Secondly, this review aggregated the results for locally advanced and metastatic pancreatic adenocarcinoma. Considering that stage three and four disease have distinct differences in characteristics, prognosis, and possibly management, it may have been more beneficial to analyse these stages separately [30]. As only published studies were included, this study was vulnerable to publication bias [31].

## 5. Conclusions

In conclusion, this systematic review confirmed that combination chemotherapy improved survival in first-line treatment of advanced pancreatic adenocarcinoma. This benefit extended to older patients. The benefit of combination chemotherapy in second-line studies for elderly patients with advanced pancreas cancer was less clear. However, the included studies showed that there may well be a difference in outcome stratified by age. Moving forward, studies need to report more comprehensive outcome data with a particular focus on providing outcome by age.

## Figures and Tables

**Figure 1 cancers-15-02289-f001:**
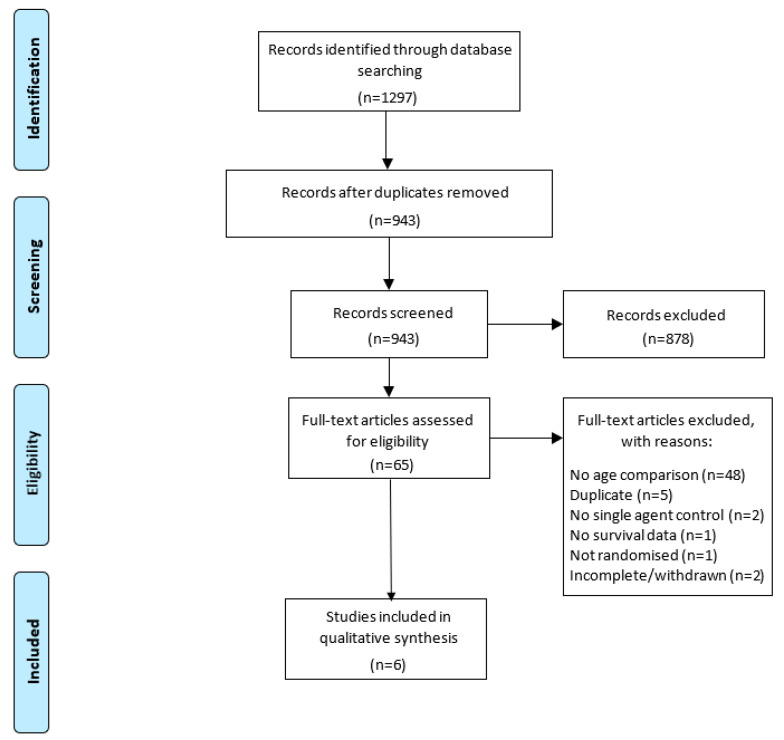
PRISMA flow diagram for the screening of articles included in this review [14].

**Figure 2 cancers-15-02289-f002:**
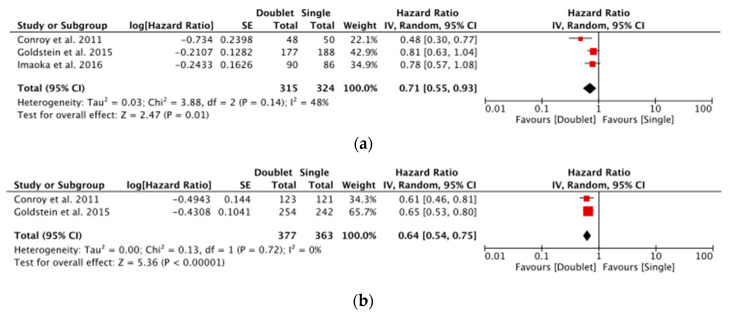
(**a**) Forest plot showing meta-analysis of overall survival in elderly patients for first-line studies [15,16,17] and (**b**) Forest plot showing meta-analysis of overall survival in young patients for first-line studies [15,16].

**Figure 3 cancers-15-02289-f003:**
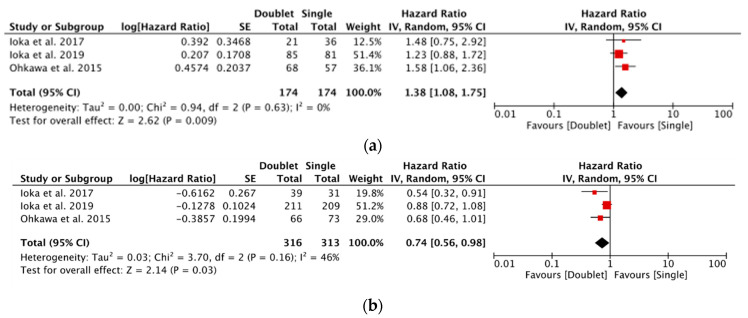
(**a**) Forest plot showing meta-analysis of overall survival in elderly patients for second-line studies [4,18,19] and (**b**) Forest plot showing meta-analysis of overall survival in young patients for second-line studies [4,18,19].

**Table 1 cancers-15-02289-t001:** Summary of studies including authors, country, treatment, number of patients overall and by subgroup, age, sex, and stage of disease.

Study	Country	n	Treatment Regimen	Patients with PDAC (n)	Mean Age (Years)	Age Range (Years)	M:F Ratio (% Male)	Locally Advanced (%)	Metastatic (%)	Elderly Subgroup (n)	Young Subgroup (n)
Conroy et al., 2011 [15]	France	171	Gemcitabine	171	61	34–75	61.4	0	100	50	121
		171	FOLFIRINOX	171	61	25–76	62	0	100	48	123
Goldstein et al., 2015 [16]	Global	430	Gemcitabine	430							
		431	Nab-Paclitaxel + Gemcitabine		63	32–88	60	0	100	189	241
Imaoka et al., 2016 [17]	Japan/ Taiwan	277	Gemcitabine	430	62	27–86	57	0	100	176	254
		280	S-1	277							
		275	Gemcitabine		-	-	61.4	23.8	76.2	86	191
Ioka et al., 2017 [18]	Japan	67	S-1	280	-	-	60.7	24.3	75.7	85	195
		60	S-1 + Irinotecan	275	-	-	57.5	24.7	75.3	90	185
Ioka et al., 2019 [19]	Japan/ Korea	290	S-1	70	65	42–76	67.2	0	100	36	31
		296	S-1 + Leucovorin	67	62	33–83	58.3	0	100	21	39
Ohkawa et al., 2015 [4]	Japan	135	S-1	302	64	32–79	57.6	0	100	81	209
		136	S-1 + Oxaliplatin	301	65	30–79	58.4	0	100	85	211

PDAC, pancreatic ductal adenocarcinoma; n, number of patients; OS, overall survival; *p*, *p* value; PFS, progression free survival; ORR, overall response rate; HR, hazard ratio; BSA, body surface area; -, data not provided.

**Table 2 cancers-15-02289-t002:** Overall outcomes of included studies including OS, PFS, and ORR along with 95% confidence intervals where provided.

Study	Country	n	Treatment Regimen	Median OS (Months)	HR	*p*	Median PFS (Months)	HR	*p*	ORR (%)	*p*
Conroy et al., 2011 [15]	France	171	Control: gemcitabine	6.8 [5.5–7.6]			3.3 [2.2–3.6]			9.4 [5.4–14.7]	
		171	Arm 1: FOLFIRINOX	11.1 [9.0–13.1]	0.57 [0.45–0.73]	<0.001	6.4 [5.5–7.2]	0.47 [0.37–0.59]	<0.001	31.6 [24.7–39.1]	<0.001
Goldstein et al., 2015 [16]	Global	430	Control: gemcitabine	6.6 [6.01–7.20]			3.7			7 [5.0–10.0]	
		431	Arm 1: gemcitabine + nab-Paclitaxel	8.7 [7.89–9.69]	0.72 [0.62–0.83]	<0.001	5.5	0.69 [0.58–0.82]	<0.001	23 [19.0–27.0]	<0.001
Imaoka et al., 2016 [17]	Japan/ Taiwan	277	Control: gemcitabine	8.8 [8.0–9.7]			4.1 [3.0–4.4]			13.3 [9.3–18.2]	NA
		280	Arm 1: S-1	9.7 [7.6–10.8]	0.96	<0.001	3.8 [2.9–4.2]	1.09	0.02	21.0 [16.1–26.6]	0.02
		275	Arm 2: S-1 + gemcitabine	10.1 [9.0–11.2]	0.88	0.15	5.7 [5.4–6.7]	0.66	<0.001	29.3 [23.7–35.5]	<0.001
Ioka et al., 2017 [18]	Japan	67	Control: S-1	5.8 [5.1–8.0]			1.9 [1.8–2.1]			6.0 [1.7–14.6]	
		60	Arm 1: S-1 + irinotecan	6.8 [5.8–9.3]	0.75 [0.51–1.09]	0.13	3.5 [2.1–4.6]	0.77 [0.53–1.11]	0.18	18.3 [9.5–30.4]	0.03
Ioka et al., 2019 [19]	Japan/ Korea	290	Control: S-1	7.9 [7.0–8.4]			2.8 [2.7–2.9]			15.1	
		296	Arm 1: S-1 + leucovorin	7.6 [7.0–8.2]	0.98 [0.82–1.16]	0.756	3.9 [2.8–4.2]	0.80 [0.67–0.95]	0.009	20.6	0.127
Ohkawa et al., 2015 [4]	Japan	135	Control: S-1	6.9 [5.8–9.0]			2.8 [1.9–3.5]			11.5 [6.6–18.3]	
		136	Arm 1: S-1 + oxaliplatin	7.4 [6.2–8.6]	NA	NA	3.0 [2.8–3.7]	NA	NA	20.9 [14.4–28.8]	0.04

n, number of patients; OS, overall survival; *p*, *p* value; PFS, progression free survival; ORR, overall response rate; HR, hazard ratio; *p*, *p* value; BSA, body surface area; NA, data not provided; [ ], 95% confidence intervals.

**Table 3 cancers-15-02289-t003:** Subgroup analysis of elderly patient outcomes including OS, PFS, and ORR along with 95% confidence intervals where provided.

Study	Treatment Arm	OS (Months)	HR	*p*	PFS (Months)	HR	*p*	ORR (%)	*p*
Conroy et al., 2011 [15]	Control: gemcitabine	NA			NA			NA	NA
	Arm 1: FOLFIRINOX	NA	0.48 [0.30–0.77]	NA	NA	NA	NA	NA	NA
Goldstein et al., 2015 [16]	Control: gemcitabine	6.5			NA			NA	NA
	Arm 1: gemcitabine + nab-Paclitaxel	7.7	0.80	0.048	NA	NA	NA	NA	NA
Imaoka et al., 2016 [17]	Control: gemcitabine	8.5 [7.4–9.4]			4.5 [3.0–5.6]			14.3 [7.1–24.7]	0.835
	Arm 1: S-1	8.0 [6.6–10.8]	0.940 [0.69–1.29]	0.715	4.2 [2.9–4.7]	1.153	0.424	25.3 [16.0–36.7]	0.309
	Arm 2: S-1 + gemcitabine	10.2 [8.8–12.4]	0.784 [0.57–1.08]	0.120	6.9 [5.6–8.3]	0.662	0.007	27.6 [18.0–39.1]	0.762
Ioka et al., 2017 [18]	Control: S-1	NA			NA			NA	NA
	Arm 1: S-1 + irinotecan	NA	1.48 [0.75–2.93]	0.07	NA	1.35 [0.70–2.60]	0.07	NA	NA
Ioka et al., 2019 [19]	Control: S-1	NA			NA			NA	NA
	Arm 1: S-1 + leucovorin	NA	1.23 [0.88–1.71]	0.093	NA	1.00 [0.72–1.38]	0.091	NA	NA
Ohkawa et al., 2015 [4]	Control: S-1	NA			NA			NA	NA
	Arm 1: S-1 plus oxaliplatin	NA	1.58 [1.06–2.36]	0.024	NA	1.07 [0.46–1.00]	0.724	NA	NA

OS, overall survival; PFS, progression free survival; ORR, objective response rate; NA, data not available; *p*, *p* value; [ ], 95% confidence intervals. Conroy et al., 2011, Goldstein et al., 2015, Ioka et al., 2017, and Ohkawa et al., 2015 used the age 65 as the age cut-off, while Imaoka et al., 2016 and Ioka et al., 2019 used 70 as the age cut-off.

**Table 4 cancers-15-02289-t004:** Subgroup analysis of young patient outcomes including OS, PFS, and ORR along with 95% confidence intervals where provided.

Study	Treatment Arm	OS (Months)	HR	*p*	PFS (Months)	HR	*p*	ORR (%)	*p*
Conroy et al., 2011 [15]	Control: gemcitabine	NA			NA			NA	NA
	Arm 1: FOLFIRINOX	NA	0.61 [0.46–0.82]	NA	NA	NA	NA	NA	NA
Goldstein et al., 2015 [16]	Control: gemcitabine	6.8			NA			NA	NA
	Arm 1: gemcitabine + nab-Paclitaxel	9.6	0.65	<0.001	NA	NA	NA	NA	NA
Imaoka et al., 2016 [17]	Control: gemcitabine	8.9 [8.1–10.0]		0.275	3.5 [2.8–4.3]		0.338	12.9 [8.2–18.8]	0.835
	Arm 1: S-1	10 [7.4–11.4]	NA	0.325	3.7 [2.9–4.2]	NA	0.691	19.1 [13.5–25.7]	0.309
	Arm 2: S-1 + gemcitabine	10.2 [8.8–12.4]	NA	0.835	5.4 [4.3–6.5]	NA	0.319	30.1 [23.3–37.7]	0.762
Ioka et al., 2017 [18]	Control: S-1	NA			NA			NA	NA
	Arm 1: S-1 + irinotecan	NA	0.54 [0.32–0.92]	0.07	NA	0.58 [0.34–0.99]	0.07	NA	NA
Ioka et al., 2019 [19]	Control: S-1	NA			NA			NA	NA
	Arm 1: S-1 + leucovorin	NA	0.88 [0.72–1.08]	0.093	NA	0.72 [0.59–0.89]	0.091	NA	NA
Ohkawa et al., 2015 [4]	Control: S-1	NA			NA			NA	NA
	Arm 1: S-1 plus oxaliplatin	NA	0.68 [0.46–1.00]	0.0498	NA	0.68 [0.47–0.98]	0.041	NA	NA

OS, overall survival; PFS, progression free survival; ORR, objective response rate; NA, data not available; *p*, *p* value; [ ], 95% confidence intervals. Conroy et al., 2011, Goldstein et al., 2015, Ioka et al., 2017, and Ohkawa et al., 2015 used the age 65 as the age cut-off, while Imaoka et al., 2016 and Ioka et al., 2019 used 70 as the age cut-off.

**Table 5 cancers-15-02289-t005:** Grade three and four adverse events for neutropenia, nausea, vomiting, and fatigue in each of the included studies.

Study	Neutropenia (%)			Nausea (%)			Vomiting (%)			Fatigue (%)		
	Single-Agent	Doublet	*p*	Single-Agent	Doublet	*p*	Single-Agent	Doublet	*p*	Single-Agent	Doublet	*p*
Conroy et al., 2011 [15]	21	45.7	<0.001	NA	NA	NA	8.3	14.5	NS	17.8	23.6	NS
Goldstein et al., 2015 [16]	26	37	NA	NA	NA	NA	NA	NA	NA	6	17	NA
Ioka et al., 2017 [18]	4.3	15.6	0.03	2.9	6.3	0.35	1.4	3.1	0.52	NA	NA	NA
Ioka et al., 2019 [19]	3.3	1.7	NA	0.7	1.7	NA	0.7	2.0	NA	0.7	1.0	NA
Ohkawa et al., 2015 [4]	11.4	8.1	NA	3.0	6.6	NA	0.8	2.9	NA	3.8	2.9	NA

NS, not significant; NA, data not available; <70, patient subgroup under 70 years old; ≥70, patient subgroup greater than or equal to 70 years old; *p*, *p* value; grey shading, first-line studies; white shading, second-line studies.

**Table 6 cancers-15-02289-t006:** Subgroup analysis by age of grade three and four adverse events for neutropenia, nausea, vomiting, and fatigue in Imaoka et al., 2016 [17].

Study	Neutropenia (%)				Nausea (%)				Vomiting (%)				Fatigue (%)			
	GEM	S-1	GEM + S-1	*p*	GEM	S-1	GEM + S-1	*p*	GEM	S-1	GEM + S-1	*p*	GEM	S-1	GEM + S-1	*p*
<70	37.2	7.7	58.9	NA	9.4	15.9	20.5	NA	5.8	7.7	16.2	NA	14.1	19.5	25.9	NA
≥70	47.7	10.6	63.3	NA	12.8	16.5	10.0	NA	9.3	10.6	7.8	NA	18.6	24.7	21.1	NA
*p*	0.113	0.487	0.513	NA	0.402	1.00	0.039	NA	0.308	0.487	0.061	NA	0.372	0.342	0.455	NA

NA, data not available; <70, patient subgroup under 70 years old; ≥70, patient subgroup greater than or equal to 70 years old; *p*, *p* value; grey shading, first-line studies; GEM, gemcitabine single-agent; S-1, S-1 single-agent; GEM + S-1, gemcitabine plus S-1 combination chemotherapy.

## Data Availability

Data are available upon request from authors.

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
