# Peer review of "Systematic Review of Single-Agent vs. Multi-Agent Chemotherapy for Advanced Pancreatic Adenocarcinoma in Elderly vs. Younger Patients"

_cancers, 2023, doi:10.3390/cancers15082289_

Round 1

Reviewer 1 Report

Dear authors, I am not convinced about the selection of the year range. Kindly re-designed the study. 

Author Response

Thank you for your review. We searched a range of 30 years. We don't believe expanding to prior to 1990 will provide clinically relevant studies nor would it change the outcome of the review.

Reviewer 2 Report

 Authors present a systematic review of randomized controlled trials comparing single-agent versus multiagent chemotherapy in advanced or metastatic pancreatic cancer.

Moreover they meta-analyzed the effect of single versus multi-agent therapy in young and elderly patients, both in first and second-line treatment.

Given the well known under-representation of elderly patients in clinical trial, although cancer predominantly affects older subjects,  I believe that the issue of the optimal treatment for this population of patients is of great practical interest.

Results of the study are very interesting from a clinical point of view. Indeed, while showing a benefit in OS for multi- versus single-agent therapy in first or second-line, the metanalysis found that for older patients single-agent treatment should be preferred in the second-line setting

As far as I can judge, the applied methodology is correct and the results clearly presented. Discussion is well written and the limitations of the study have been correctly addressed. In particular, as most of the included studies used S-1 which is commonly used in Far East Countries,  the Authors aknowledged the fact that results cannot be directly extrapolated for Western ones

However, that does not alter the fact that the paper addresses an issue of general clinical interest and, in my opinion,  is certainly eligible for publication.

Author Response

We thank you for your review. We also hope that this study will be of clinical use.

Reviewer 3 Report

The manuscript entitled  Systematic review of single-agent vs multi agent chemotherapy 2
for advanced pancreatic adenocarcinoma in elderly vs younger 3
patientsby Lewis and Nagrial is well designed study.

The best part of the manuscript is its discussion and authors clearly mentioned the limitations as well.

The quality of some figures should be improved.

Author Response

We thank you for your comments. We will improve the tables and improve the definition of the figures.

Round 2

Reviewer 1 Report

Please highlight the reason for the time range set in the introduction section.  

Author Response

We have updated the methods to highlight that the time frame of 1990 - 2020 was selected as the first study demonstrating a survival benefit of chemotherapy in pancreas cancer wasn't published until 1997 (Burris et al.)